# Vibration Analysis of a Unimorph Nanobeam with a Dielectric Layer of Both Flexoelectricity and Piezoelectricity

**DOI:** 10.3390/ma16093485

**Published:** 2023-04-30

**Authors:** Ali Naderi, Tran Quoc-Thai, Xiaoying Zhuang, Xiaoning Jiang

**Affiliations:** 1Department of Mechanical and Aerospace Engineering, Syracuse University, Syracuse, NY 13244, USA; 2Fluid Dynamics and Solid Mechanics, T-3, Theoretical Division, Los Alamos National Laboratory, Los Alamos, NM 87545, USA; 3Department of Mathematics and Physics, Leibniz Universität Hannover, 30419 Hanover, Germany; 4Department of Mechanical and Aerospace Engineering, North Carolina State University, Raleigh, NC 27695, USA

**Keywords:** flexoelectric, two-phase local/nonlocal elasticity, energy harvesting, vibration, GDQM

## Abstract

In this study, for the first time, free and forced vibrational responses of a unimorph nanobeam consisting of a functionally graded base, along with a dielectric layer of both piezoelectricity and flexoelectricity, is investigated based on paradox-free local/nonlocal elasticity. The formulation and boundary conditions are attained by utilizing the energy method Hamilton’s principle. In order to set a comparison, the formulation of a model in the framework of differential nonlocal is first presented. An effective implementation of the generalized differential quadrature method (GDQM) is then utilized to solve higher-order partial differential equations. This method can be utilized to solve the complex equations whose analytic results are quite difficult to obtain. Lastly, the impact of various parameters is studied to characterize the vibrational behavior of the system. Additionally, the major impact of flexoelectricity compared to piezoelectricity on a small scale is exhibited. The results show that small-scale flexoelectricity, rather than piezoelectricity, is dominant in electromechanical coupling. One of the results that can be mentioned is that the beams with higher nonlocality have the higher voltage and displacement under the same excitation amplitude. The findings can be helpful for further theoretical as well as experimental studies in which dielectric material is used in smart structures.

## 1. Introduction

The big data era demands developments in micro/nano-electromechanical systems (M/NEMS) including nanosensors [1,2], nanotransistors [3], nanoharvesters [4,5], and nanoelectronics. The electromechanical coupling in dielectric materials plays a crucial role in ascertaining the performance of many M/NEMS devices. The electromechanical coupling was mainly related to piezoelectricity resulting from a uniform strain of the dielectric materials. However, recently, the coupling effect due to the nonuniform strain, i.e., strain gradient, has become increasingly attractive to researchers in exploring the potential of small-scale smart structures, since the gradient of the strain becomes more predominant with reduced sizes, especially at a sub-micron and nanoscale [6,7]. Thus, to capture the size effect, as the classic continuum theories are insufficient, a large amount of effort has been devoted to enriching the continuum theories by introducing an additional material length scale. The integral nonlocal formulation accounts for the size effect [8,9], which assumes that the strain in a particular point of a structure results not only from local forces applied at that point but also from other forces applied to other regions of the domain. In the effort of enhancing the conventional elastic theory, the differential form of nonlocal theory [10], largely due to less complexity together with a reduction in computational costs compared to the integral form, has been proposed. Based on the differential form of nonlocal elasticity, vibrations [11,12], critical buckling loads [13], and wave propagation [14,15] associated with small-scaled structures have been reported. By using nonlocal elasticity together with higher order beam theory, Pham et al. [16] presented an article studying the vibrational response of FG curved beam made of porous materials. The beam was subjected to hygro-thermal loading. Additionally, the vibration analysis associated with nanobeams placed on elastic substrate were modeled via nonlocal strain gradient theory were conducted using FEM [17]. However, it was later discovered that this theory has some inconsistencies and was unable to yield reliable results [18]. Subsequently, a large number of investigations have been conducted to verify this issue [19]. The main problem with this approach is the inability to capture the softening effect by applying the nonlocality, specifically in the nanobeams with clamped-free end conditions [20]. Later, it was found that this problem is due to a lack of additional boundary conditions in the transformation process from the integral form to the differential form [21]. Moreover, it should be noted that the studies on lower-scale structures which are modeled via integral nonlocal revealed that this model has no inconsistencies [22,23]. There have been quite a few studies conducted in order to mend the inconsistencies of differential nonlocal, such as the stress-driven model presented by Romano Barretta [24] and employing two-phase local/nonlocal elasticity, which basically includes the nonlocal integral theory in addition to a classic part. The credibility of this theory was confirmed by molecular dynamics as well as experimental studies [25]. Similar to the nonlocal theory, the two-phase theory is more applicable provided that it is in a differential form. In contrast to differential nonlocal, the transformation associated with two-phase local/nonlocal elasticity, by introducing two additional boundary conditions, has no inconsistencies in investigating the behavior of small-scale structures. Additionally, it has been shown that the two-phase theory is capable of capturing the softening effect due to nonlocality. Because of this reason, there are many articles in which two-phase theory has been employed to capture the size effect, including vibration response [26] and buckling [27] of nanobeams based on Euler–Bernoulli beam theory. Based on the two-phase theory, Fakher et al. [28] have explored the vibration alongside the buckling of a nanobeam. In their work, they presented the effect of size-dependency on the thermal load as well as foundation loads. Behdad et al. [29] investigated the vibrational characteristics related to defected nanobeams that are placed on a two-parameter type of elastic medium. Additionally, in the framework of two-phase elasticity, the dynamic stability associated with nanobeams made of functionally graded porous media under mechanical loading was explored [30]. Using two-phase elasticity, Selvamani et al. [31] investigated the deformation associated with nanobeams made of graphene oxide powder composites. The nanobeam was under thermal and electrical loading. Additionally, Hosseini-Hashemi et al. [32] investigated the vibration of viscoelastic Euler–Bernoulli nanobeams with considering surface effect. Lately, by introducing a new approach of using GDQM, Naderi et al. [33] managed to present a paper on the vibrational behavior of magneto–piezo nanobeams which are resting on a viscoelastic foundation. To construct the C1 continuous scheme, isogeometric analysis (IGA) has been widely applied for flexoelectric and strain gradient effects [34,35,36]. However, similar to FEM, IGA is a local approximation method and it requires a large number of control points to guarantee the result’s accuracy. GDQM, on the other hand, is a global approximation method employing directly the hermit functions allowing us to evaluate the strain gradient naturally without any recursion procedure to construct higher-order shape functions. Thus, although two-phase elasticity increases the complexity of the formulation, using this elasticity results in much more accurate results without any paradoxes.

The flexoelectric effect, as one of the electromechanical couplings in dielectric materials due to nonuniform strain in the structures, has been the topic of various studies. In this regard, the impact of flexoelectricity on the vibrational characteristics of nanobeams was studied by Nguyen et al. [34]. There are quite a few articles that have utilized the strain gradient effect of flexoelectric material for sensing and energy harvesting [37,38]. Jiang et al. [39] reviewed the flexoelectricity in various materials as well as the application of flexoelectricity in sensors and actuators. Additionally, using the strain gradient sensing of flexoelectric materials, a sensor for detecting crack growth was presented [7]. Further, the enhancement impact of considering the flexoelectricity—resulting from nonuniform strain—on the energy harvesting of piezo as well as non-piezo materials was investigated [40]. Yan and Jiang [41] presented a study to show the effect of flexoelectricity on the bending of nanobeams under electrical as well as mechanical loading. In another paper, by using Timoshenko beam theory, they explored the flexoelectricity impacts on the dynamic and static responses of simply supported nanobeams [42]. In addition, the nonlocal theory has been used to capture the size effect. For instance, Sidhardh and Ray [43], by employing the finite element method, examined the static bending of a two-layered nanobeam, including a layer with a flexoelectric effect as an actuator. In their report, elasticity was utilized as the size-dependent theory. Additionally, based on strain gradient theory and isotropic flexoelectric theory, the vibration analysis of microplates considering the microscopic electrical field, polarization gradient, and strain gradient effects were examined [44]. In these studies, the extensive flexoelectric characteristics on the system’s dynamic and static responses was exhibited.

In this work, the free and forced vibration, as well as the output voltage of a unimorph nanobeam including a functionally graded (FG) base along with a piezo–flexoelectric dielectric layer, is modeled for the first time, according to paradox-free elasticity or two-phase local/nonlocal theory. The formulation and boundary conditions are extracted by utilizing the energy method, i.e., Hamilton’s principle. The nonlocal differential elastic beam formulations are presented and the GDQM is employed. The performance of the proposed model is presented by studying the influence of the input material parameters, boundary conditions, and structural dimensions. The obtained numerical results indicate the possibility of our proposed approach in characterizing the vibrational behavior of the piezo–flexoelectric bimorph nanobeam.

## 2. Problem Formulation

The vibration of a unimorph nanobeam energy harvesting including a dielectric layer with the flexoelectricity and piezoelectricity as well as a functionally graded (FG) base is examined. The Euler beam theory as well as two-phase local/nonlocal elasticity are employed. Figure 1 exhibits the schematic view for the nanobeam with *L, b, h,* and *h_d_* as its length, width, the thickness of FG base, and thickness of the dielectric layer, respectively. Open electric circuit condition is applied, and the voltage difference between the upper and lower dielectric layers is measured.

Here, the electrical enthalpy for the dielectric layer with both the piezoelectricity and flexoelectricity is as follows [45]
(1)$$\begin{array}{l} H=\frac{1}{2}c_{ijkl}\varepsilon_{ij}\varepsilon_{kl}-\frac{1}{2}\kappa_{ij}E_{i}E_{j}-\frac{1}{2}b_{ijkl}E_{k,l}E_{i,j}\\ -\mu_{ijkl}E_{k}\varepsilon_{ij,l}-e_{ijk}E_{i}\varepsilon_{jk}+\mu_{ijkl}E_{k,l}\varepsilon_{ij}, \end{array}$$
in which $c_{ijkl}$ denotes the elastic constant and the dielectric constant tensor is $\kappa_{kl}$. Additionally, $\mu_{ijkl}$, $b_{ijkl}$ as well as $e_{kij}$ are flexoelectric constants, nonlocal electrical coupling, and piezoelectric constants, respectively. In addition, $E_{l}$ and $E_{k}$ are electrical fields, and $\varepsilon_{ij}$ represent the strain tensor of the beam.

Now, according to Euler–Bernoulli theory, the displacement field for the neutral axis of the nanobeam is as follows [26]
(2)$$\begin{array}{l} \;\;U_{x}\left(x,z,t\right)=-z\;\frac{\partial w\left(x,t\right)}{\partial x}\;,\;\;\;\;\\ U_{z}\left(x,z,t\right)=w\left(x,t\right), \end{array}$$
in which $U_{x}$ and $U_{z}$ are displacement fields in *x* and *z* directions. Additionally, $w\left(x,t\right)$ and $\frac{\partial w\left(x,t\right)}{\partial x}$ are transverse deflection and rotation of the neutral plane of the beam. Further, the corresponding strains of the nanobeam are [26]
(3)$$\begin{array}{l} \varepsilon_{xx}=-z\frac{\partial^{2}w\left(x,t\right)}{\partial x^{2}}\;,\\ \varepsilon_{xx,z}=-\frac{\partial^{2}w\left(x,t\right)}{\partial x^{2}}, \end{array}$$
where $\varepsilon_{xx}$ and $\varepsilon_{xx,z}$ show the elastic strain and the gradient of the strain, respectively.

Here, the stress–strain relations associated with the functionally graded base is as follows [46]
(4)$$\sigma^{b}{}_{xx}=E\left(z\right)\varepsilon_{xx},$$
where $E\left(z\right)$ is elastic modulus associated with the FG layer and $\sigma^{b}{}_{xx}$ is the stress tensor related to the base layer. Next, the following equation represents the normal and shear stress associated with piezoelectric nanobeams with the flexoelectric effect considered [47].
(5)$$\begin{array}{l} \sigma^{d}{}_{xx}=c_{11}\varepsilon_{xx}-e_{31}E_{z}+\mu_{1133}E_{z,z},\\ \tau_{xxz}=-\mu_{1133}E_{z}, \end{array}$$
where $E_{z}=-\frac{\partial \phi }{\partial z}$ and ϕ is electrical potential, and $\sigma^{d}{}_{xx}$ as well as $\tau_{xxz}$ represent the normal and second-order stress in the dielectric layer. Additionally, superscript *d* shows the dielectric layer. The dielectric nanobeam’s electrical displacement and quadrupolar contribution are as follows [47].
(6)$$\begin{array}{l} D_{z}=\kappa_{33}E_{z}+e_{31}\varepsilon_{xx}+\mu_{1133}\varepsilon_{xx,z},\\ Q_{zz}=b_{33}E_{z,z}-\mu_{1133}\varepsilon_{xx}. \end{array}$$
in which $D_{z}$ and $Q_{zz}$ are electrical displacement and quadrupolar contribution. Now, with free electrical charges being zero, the following can be written based on Gauss’s law [48].
(7)$$\frac{\partial D_{z}}{\partial z}-\frac{\partial^{2}Q_{zz}}{\partial z^{2}}=0.$$

Here, by using Equation (6) into Equation (7), the equation below can be yielded.
(8)$$\kappa_{33}E_{z,z}+e_{31}\varepsilon_{xx,z}-b_{33}E_{z,zzz}+2\mu_{1133}\varepsilon_{xx,zz}=0.$$

In addition, by considering the following boundary conditions for electrical potential
(9)$$\phi \left(\frac{h}{2}\right)=0\;\;,\;\;\phi \left(\frac{h}{2}+h_{d}\right)=V,\;\;Q_{zz}\left(\frac{h}{2}\right)=0,Q_{zz}\left(\frac{h}{2}+h_{d}\right)=0$$

Equation (8) can be solved for electrical potential as follows [48,49]
(10)$$\begin{array}{l} \phi \left(z\right)=\frac{1}{8}\left(-\frac{4hV}{h_{d}}+\frac{\partial^{2}w}{\partial x^{2}}\frac{e_{31}\left(-8-h\left(h+2h_{d}\right)\eta^{2}\right)}{b_{33}\eta^{4}}\right)\\ \frac{V\;z}{h_{d}}-\frac{z^{2}e_{31}}{2\kappa_{33}}\frac{\partial^{2}w}{\partial x^{2}}+\frac{1}{2}z\left(\frac{e_{31}\left(h+h_{d}\right)}{\kappa_{33}}-\frac{2\mu_{1133}}{\eta^{2}b_{33}}\right)\frac{\partial^{2}w}{\partial x^{2}}+\\ \frac{e^{-\frac{1}{2}\left(h+2z\right)\eta }\left(-1+\coth\left[\eta h_{d}\right]\right)}{4\eta^{2}b_{33}\kappa_{33}}\frac{\partial^{2}w}{\partial x^{2}}\left(\begin{array}{l} 2\left(-1+e^{\eta h_{d}}\right)\left(e^{2z\eta }+e^{\eta \left(h+h_{d}\right)}\right)b_{33}e_{31}\\ -\left(\begin{array}{l} -\left(\left(-1+e^{\eta h_{d}}\right)\left(e^{2z\eta }+e^{\eta \left(h+h_{d}\right)}\right)h\right)\\ +2e^{\eta h_{d}}\left(e^{h\eta }-e^{2z\eta }\right)h_{d} \end{array}\right)\kappa_{33}\mu_{1133} \end{array}\right) \end{array}$$
where $\eta =\sqrt{\frac{\kappa_{33}}{b_{33}}}$ based on Equation (10), $E_{z}$ and $E_{z,z}$ can be obtained as follows.
(11)$$\begin{array}{l} E_{z}=\frac{V}{h_{d}}-\frac{ze_{31}}{\kappa_{33}}\frac{\partial^{2}w}{\partial x^{2}}+\frac{1}{2}\left(\frac{e_{31}\left(h+h_{d}\right)}{\kappa_{33}}-\frac{2\mu_{1133}}{\eta^{2}b_{33}}\right)\frac{\partial^{2}w}{\partial x^{2}}-\\ \frac{e^{-\frac{1}{2}\left(h+2z\right)\eta }\left(-1+\coth\left[\eta h_{d}\right]\right)}{4\eta b_{33}\kappa_{33}}\frac{\partial^{2}w}{\partial x^{2}}\left(\begin{array}{l} 2\left(-1+e^{\eta h_{d}}\right)\left(3e^{2z\eta }+e^{\eta \left(h+h_{d}\right)}\right)b_{33}e_{31}\\ -\left(\begin{array}{l} -\left(\left(-1+e^{\eta h_{d}}\right)\left(3e^{2z\eta }+e^{\eta \left(h+h_{d}\right)}\right)h\right)\\ +2e^{\eta h_{d}}\left(e^{h\eta }-3e^{2z\eta }\right)h_{d} \end{array}\right)\kappa_{33}\mu_{1133} \end{array}\right)\\ E_{z,z}=\frac{e^{-\frac{1}{2}\left(h+2z\right)\eta }}{2\left(-1+e^{2\eta h_{d}}\right)b_{33}\kappa_{33}}\left(\begin{array}{l} 2\left(e^{\frac{h\eta }{2}}-e^{z\eta }\right)\left(-1+e^{\eta h_{d}}\right)\left(-e^{z\eta }+e^{\frac{h\eta }{2}+\eta h_{d}}\right)b_{33}e_{31}\\ -\left(\begin{array}{l} -\left(\left(-1+e^{\eta h_{d}}\right)\left(e^{2z\eta }+e^{\eta \left(h+h_{d}\right)}\right)h\right)\\ +2e^{\eta h_{d}}\left(e^{h\eta }-e^{2z\eta }\right)h_{d} \end{array}\right)\kappa_{33}\mu_{1133} \end{array}\right)\frac{\partial^{2}w}{\partial x^{2}} \end{array}$$

Additionally, the functionally graded material properties used in the base of the beam are
(12)$$\begin{array}{l} \rho \left(z\right)=\rho_{m}+\left(\rho_{cm}-\rho_{m}\right){\left(\frac{z}{h}+\frac{1}{2}\right)}^{n},\\ E\left(z\right)=E_{m}+\left(E_{cm}-E_{m}\right){\left(\frac{z}{h}+\frac{1}{2}\right)}^{n},\\ \nu \left(z\right)=\nu_{m}+\left(\nu_{cm}-\nu_{m}\right){\left(\frac{z}{h}+\frac{1}{2}\right)}^{n}, \end{array}$$
in which $\rho_{cm}$ and $\rho_{m}$ denote the mass density related to metal and ceramic in the FG layer, respectively, and $E_{cm}$ and *E_m_* represent the ceramic and metal elastic modulus, respectively. Additionally, $\nu_{m}$ and $\nu_{cm}$ are Poisson’s ratio of metal and ceramic, and *n* represents the FG power index.

Now, the strain $\Pi_{s}$ and kinetic $\Pi_{k}$ energy of the nanobeam made of two layers, including a dielectric layer, considering the flexoelectricity and piezoelectricity and an elastic base are shown in the following equation.
(13)$$\begin{array}{l} \Pi_{s}=\frac{1}{2}b\int_{0}^{L}\int_{-h/2}^{h/2}\left(\sigma^{b}{}_{xx}\varepsilon_{xx}\right)dz\;dx+\frac{1}{2}b\int_{0}^{L}\int_{h/2}^{h/2+h_{d}}\left(\sigma^{d}{}_{xx}\varepsilon_{xx}+\tau_{xxz}\varepsilon_{xx,z}\right)dz\;dx,\\ \Pi_{k}=\frac{1}{2}\int_{0}^{L}I_{1}{\left(\frac{\partial w\left(x,t\right)}{\partial t}\right)}^{2}\;dx, \end{array}$$
where $I_{1}=\int_{-h/2}^{h/2}\rho \left(z\right)dz+\int_{h/2}^{h/2+h_{d}}\rho^{d}dz$.

Here, the energy method is utilized to attain the equations in addition to geometrical end conditions.
(14)$$\int_{0}^{t}\delta \left(\Pi_{s}-\Pi_{k}\right)dt=0$$

The following equation, representing the governing equation associated with the vibration of a two-layered nanobeam that contains an FG base and a dielectric layer can be achieved using Equation (14).
(15)$$\begin{array}{l} \delta \phi :\;\int_{A}\frac{\partial D_{z}}{\partial z}dA=0\\ \delta w:\frac{\partial^{2}P\left(x,t\right)}{\partial x^{2}}+\;\frac{\partial^{2}M\left(x,t\right)}{\partial x^{2}}-\rho A\frac{\partial^{2}w\left(x,t\right)}{\partial t^{2}}=0 \end{array}$$
in which
(16)$$\begin{array}{l} M\left(x,t\right)=b\int_{-h/2}^{h/2}z\sigma_{xx}^{b}dz+b\int_{h/2}^{h/2+h_{d}}z\sigma_{xx}^{d}dz,\\ P\left(x,t\right)=b\int_{h/2}^{h/2+h_{d}}\tau_{xxz}dzb. \end{array}$$

Moreover, the end conditions are
(17)$$\begin{array}{l} w=0\;\;\;\;\;\;or\;\;\;\;\;\;V\left(x,t\right)\equiv \frac{\partial M}{\partial x}+\frac{\partial P}{\partial x}=0,\\ \frac{\partial w}{\partial x}=0\;\;\;\;\;\;or\;\;\;\;\;\;M+P=0.\\ \phi =0\;\;\;\;\;\;\;\;or\;\;\;\;\;\;\;\;Q_{zz}n_{z}=0 \end{array}$$

### Two-Phase Local/Nonlocal Theory

The stress–strain relation defined in the two-phase local/nonlocal theory is [26]
(18)$$\begin{array}{l} t\left(x\right)\;=\xi \;C:\varepsilon \left(x\right)\;+\;\left(1-\xi \right)\int_{V}\alpha \left(x,\bar{x},\kappa \right)\;C:\varepsilon \left(\bar{x}\right)\;d\bar{V}\\ where\\ \alpha \left(x,\bar{x},\kappa \right)\;=\;\frac{1}{2\;k}\;e^{-\;\frac{\left|x-\bar{x}\right|}{k}}, \end{array}$$
in which C, $t\left(x\right)$, $\varepsilon \left(x\right)$, x, $\alpha \left(x,\bar{x},\kappa \right)$, ξ, V, and k denote the fourth-order elasticity tensor, Cauchy stress tensor in the two-phase state, the strain tensor, reference point, kernel function, local phase fraction factor, domain volume, and nonlocal parameter, respectively. Thus, two-phase elasticity can be written as follows
(19)$$Q\left(x,t\right)=\xi T\left(x,t\right)+\left(\frac{1-\xi }{2k}\right)\int_{0}^{L}e^{-\frac{\left|x-\bar{x}\right|}{k}}T\left(\bar{x},t\right)d\bar{x}.$$
in which $T\left(x,t\right)$ is a function of the local quadrupolar contribution. It should be mentioned that by setting ξ=0 in Equation (18), the pure nonlocal theory can be attained. Now, by utilizing the transformation presented by Polyanin et al. [50], which was firstly used for two-phase elasticity by Fernandez et al. [26], the integral formulation is written in the differential format as follows
(20)$$-T\left(x,t\right)+\xi k^{2}\frac{\partial^{2}T\left(x,t\right)}{\partial x^{2}}+Q\left(x,t\right)-k^{2}\frac{\partial^{2}Q\left(x,t\right)}{\partial x^{2}}=0.$$

Here, it is vital to mention that this transformation can be used only when two constitutive boundary conditions—CBCs—are satisfied. Therefore, the CBCs at the tips of the nanobeam are given.
(21)$$\begin{array}{l} \frac{Q\left(x,t\right)}{k}-\frac{\partial Q\left(x,t\right)}{\partial x}+\xi \frac{\partial T\left(x,t\right)}{\partial x}-\frac{\xi }{k}T\left(x,t\right)=0\;\;\;\;\;\;\;At\;\;\;x=0,\\ -\frac{Q\left(x,t\right)}{k}-\frac{\partial Q\left(x,t\right)}{\partial x}\;+\xi \frac{\partial T\left(x,t\right)}{\partial x}+\frac{\xi }{k}T\left(x,t\right)=0\;\;\;\;\;\;At\;\;\;x=L. \end{array}$$

Now, the stress–strain relations and the electrical displacements can be rewritten in two-phase form using Equation (18) [43,51].
(22)$$\begin{array}{l} \sigma^{d}{}_{xx}=\xi \left(c_{11}\varepsilon_{xx}-e_{31}E_{z}+\mu_{1133}E_{z,z}\right)+\left(\frac{1-\xi }{2k}\right)\int_{0}^{L}e^{-\frac{\left|x-\bar{x}\right|}{k}}\left(c_{11}\varepsilon_{xx}-e_{31}E_{z}+\mu_{1133}E_{z,z}\right)d\bar{x},\\ \sigma^{b}{}_{xx}=\xi \left(E\left(z\right)\varepsilon_{xx}\right)+\left(\frac{1-\xi }{2k}\right)\int_{0}^{L}e^{-\frac{\left|x-\bar{x}\right|}{k}}\left(E\left(z\right)\varepsilon_{xx}\right)d\bar{x},\\ \tau_{xxz}=\xi \left(-\mu_{1133}E_{z}\right)+\left(\frac{1-\xi }{2k}\right)\int_{0}^{L}e^{-\frac{\left|x-\bar{x}\right|}{k}}\left(-\mu_{1133}E_{z}\right)d\bar{x}. \end{array}$$

Now—by using Equations (21)–(22)—the two-phase bending moment, two-phase higher-order bending moment in differential form are derived and presented in Section A.1 and Section A.2.

## 3. Solution Procedure

### Generalized Differential Quadrature Method

In this section, the solution procedure that is utilized to extract the forced and free vibrational response of a nanobeam which is made of two layers, including an FG base and a layer made of dielectric material considering flexoelectric and piezoelectric effects, is introduced. The GDQM from [33] is used. Based on this method, the momentum and higher-order momentum in addition to the deflection are the independent variables which can be rewritten using the separation of variables as follows.
(23)$$\left\{\begin{array}{c} w\left(x,t\right)\\ M\left(x,t\right)\\ P\left(x,t\right) \end{array}\right\}=\left\{\begin{array}{c} W\left(x\right)\\ \bar{M}\left(x\right)\\ \bar{P}\left(x\right) \end{array}\right\}\;T\left(t\right)$$

Now, by employing the fourth-order GDQM, the *n*-th derivative of any function such as $ℑ\left(x\right)$ can be written as follows.
(24)$${ℑ}^{\left(r\right)}\left(x_{i}\right)=\sum_{j=1}^{ns}\Lambda_{j0}^{\left(r\right)}\left(x_{i}\right){ℑ}_{j}+\Lambda_{1\;1}^{\left(r\right)}\left(x_{i}\right){ℑ}_{1}^{\left(1\right)}+\Lambda_{ns\;1}^{\left(r\right)}\left(x_{i}\right){ℑ}_{ns}^{\left(1\right)}=\;\sum_{j=1}^{ns+2}\Lambda_{ij}^{\left(r\right)}{℘}_{j}\;\;\;\;\;for\;\;i=1,2,3,\ldots ns-1,\;ns$$

In which $\Lambda_{jl}^{\left(r\right)}\left(x_{i}\right)$ is Hermit interpolation, and ns denotes the number of sampling points. Additionally, the location corresponded to the grid points is $x_{i}$. Here, the discrete form of the momentum, higher-order momentum, and displacement field can be given as the following equation together with Figure 2.
(25)$${\left\{{℘}_{j}\right\}}^{T}={\left\{W_{1},\ldots ,W_{ns+2},M_{1},\ldots ,M_{ns+2},P_{1},\ldots ,P_{ns+2}\right\}}^{T}$$

In which $M_{j}$, $P_{j}$, $W_{j}$ are the discretized form of bending moment, higher-order bending moment, and lateral displacement.

**Figure 2 materials-16-03485-f002:**
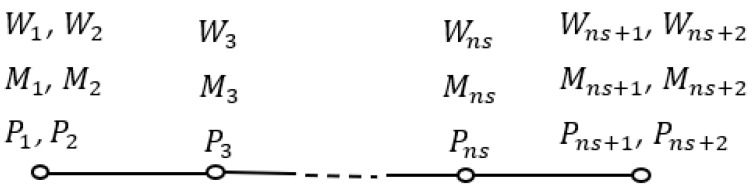
The schematic of the distribution of grid points on the nanobeam.

Where the subscripts 2 and *ns +* 2 are the derivative of the variables at the borders, 1 and *ns +* 1 are the variable values at the border points, and points 3, …, *ns* represent the domain values of the variables.

Additionally, the distribution of grid points is selected by the following method.
(26)$$x_{i}=L\frac{\left(1-\cos\left(\frac{\left(i-1\right)}{\left(ns-1\right)}\pi \right)\right)\;}{2}\;\;\;\;\;\;\;\;\;for\;\;i=1,2,3,\ldots ns-1,\;ns.$$

Now, the Hermit interpolation can be introduced as
(27)$$\Lambda_{jl}^{\left(r\right)}\left(x_{i}\right)\;=\;\left\{\begin{array}{l} 1\;\;\;\;\;\;\;\;\;\;\;\;\;\;\mathrm{if}\;\;i=j\;\;\&\;l=r\\ 0\;\;\;\;\;\;\;\;\;\;\;\;\;\;\mathrm{otherwise}\; \end{array}\right.\;\;\;\;\;$$
where
(28)$$\begin{array}{l} \Lambda_{j0}\left(x\right)=\frac{\left(x-x_{1}\right)\left(x-x_{ns}\right)}{\left(x_{j}-x_{1}\right)\left(x_{j}-x_{ns}\right)}E_{j}\left(x\right)\;\;\;\;\;\;\;for\;\;j=2,3,\ldots ,\;ns-1.\\ \;\Lambda_{ri}\left(x\right)=\left(c_{ri}+b_{ri}x+a_{ri}x^{2}\right)E_{r}\left(x\right)\;\;\;\;\;\;\;\;\;for\;\;r=1,\mathrm{ns}\;\;\;\;\;\;\mathrm{and}\;\;\;\;\;i=0,1 \end{array}$$
where $E_{j}\left(x_{i}\right)$ are the Lagrange interpolation and the constant, $a_{ri}$, $b_{ri}$, and $c_{ri}$, can be found in Section A.3.

Additionally, the Lagrange interpolation can be written as
(29)$$E_{j}^{\left(1\right)}\left(x_{i}\right)=\left\{\begin{array}{l} \frac{Z^{\left(1\right)}\left(x_{i}\right)}{\left(x_{i}-x_{j}\right)Z^{\left(1\right)}\left(x_{j}\right)}\;\;\;\;\;\;\;\;\;\;\;\;\;\;\;\;\;\;\;\;\;\;\;\;\;\;\;\;\;\;\;\;\;\;\;\;\;\;\;for\;\;\;\;\;i,j=1,2,3,\ldots ,ns;\;\;\;i\neq j\\ -\sum_{j=1,i\neq j}^{ng}E_{j}^{\left(1\right)}\left(x_{i}\right)\;\;\;\;\;\;\;\;\;\;\;\;\;\;\;\;\;\;\;\;\;\;\;\;\;\;\;\;\;\;\;\;\;\;\;\;\;\;\;\;\;\;\;\;\;\;\;\;for\;\;\;\;\;i,j=1,2,3,\ldots ,ns \end{array}\right.$$
where $Z^{\left(1\right)}\left(x_{i}\right)=\prod_{m=1,m\neq i}^{ns}\left(x_{i}-x_{m}\right)$. Additionally, the higher-order derivative associated with the Lagrange interpolation can be obtained in the following equation.
(30)$$E_{j}^{\left(r\right)}\left(x_{i}\right)=\left\{\begin{array}{l} r\left(E_{i}^{\left(r-1\right)}\left(x_{i}\right)E_{j}^{\left(1\right)}\left(x_{i}\right)-\frac{E_{j}^{\left(r-1\right)}\left(x_{i}\right)}{\left(x_{i}-x_{j}\right)}\right)\;\;\;\;\;\;\;\;\;\;\;\;\;\;\;\;\;\;\;\;\;\left(\mathrm{for}\;i,j=1,2,\ldots ,ns,\;i\neq j\right)\\ -\sum_{j=1,i\neq j}^{ns}E_{j}^{\left(r\right)}\left(x_{i}\right)\;\;\;\;\;\;\;\;\;\;\;\;\;\;\;\;\;\;\;\;\;\;\;\;\;\;\;\;\;\;\;\;\;\;\;\;\;\;\left(\mathrm{for}\;i,j=1,2,\ldots ,ns\;,\;i=j\right) \end{array}\right.$$

Now, the discretized formulation and boundary conditions presented in Section A.4 can be written in a matrix format as follows.
(31)$$\begin{array}{l} \left[\begin{array}{cc} {\left[K_{bb}\right]}_{12\times 12} & {\left[K_{bd}\right]}_{12\times \left(3ns-6\right)}\\ {\left[K_{db}\right]}_{\left(3ns-6\right)\times 12} & {\left[K_{dd}\right]}_{\left(3ns-6\right)\times \left(3ns-6\right)} \end{array}\right].\left\{\begin{array}{c} \left\{V_{b}\right\}\\ \left\{V_{d}\right\} \end{array}\right\}\\ -\omega^{2}\left[\begin{array}{cc} {\left[M_{bb}\right]}_{12\times 12} & {\left[M_{bd}\right]}_{12\times \left(3ns-6\right)}\\ {\left[M_{db}\right]}_{\left(3ns-6\right)\times 12} & {\left[M_{dd}\right]}_{\left(3ns-6\right)\times \left(3ns-6\right)} \end{array}\right].\left\{\begin{array}{c} \left\{V_{b}\right\}\\ \left\{V_{d}\right\} \end{array}\right\}=0. \end{array}$$
in which *b* and *d* show boundary and domain grid points, respectively. Additionally, the discretized form of formulation associated with the differential form of the purely nonlocal model is presented in Section A.5. Next, the forced vibration analysis results are obtained using Newmark-beta and GIQM procedure, as explained in Ref. [1].

Additionally, the damping matrix is
(32)$$C_{ww}=\beta \left[K\right]+\alpha \left[M\right]$$
where
(33)$$\begin{array}{l} \alpha =2\xi_{i}\left(\frac{\omega_{i}\omega_{j}}{\omega_{i}+\omega_{j}}\right)\\ \beta =2\xi_{i}\left(\frac{1}{\omega_{i}+\omega_{j}}\right) \end{array}$$
where $\xi_{i}$ is the damping ratio and $\omega_{i}$ as well as $\omega_{j}$ are the natural frequencies.

In addition, the harmonic force applied to the system for the forced vibrational results can be defined as follows.
(34)$$F^{ext}={\bar{f}}_{0}\sin\left(\Omega t\right)$$
in which ${\bar{f}}_{0}$ and Ω denote the amplitude and the frequency of the force.

## 4. Numerical Results and Discussion

This section presents the numerical results related to the vibration of a unimorph nanobeam, which is made of a dielectric layer with consideration of flexoelectric and piezoelectric effects as well as an FG layer based on two-phase elasticity. It should be mentioned that, in this study, it is supposed that the nanobeam is made of BaTiO_3_ with material properties such as $c_{11}=167.55\;\left(\mathrm{GPa}\right)$, $\rho =6020\;\left(\mathrm{kg}/m^{3}\right)$, $e_{31}=4.43\left(C/m^{2}\right)\;$,$b_{33}=1.265\times 10^{-17}\left({\mathrm{Jm}}^{2}/\mathrm{VC}\right)$, $\kappa_{33}=1.265\times 10^{-8}\left(C/\mathrm{Vm}\right)$, and $\mu_{1133}=5\times 10^{-8}\left(C/m\right)$ [52]. Additionally, the mechanical properties related to the FG layer are $\nu_{m}=0.3$, $\nu_{cm}=0.2$, $E_{m}=70\left(GPa\right)$, $E_{cm}=200\left(GPa\right)$, $\rho_{m}=2702\;\left(\mathrm{kg}/m^{3}\right)$, and $\rho_{cm}=5700\;\left(\mathrm{kg}/m^{3}\right)$. Additionally, the damping ratio is $\xi_{i}=0.05$ and $b=h+h_{d}$. It should be mentioned that from [52], the value of flexoelectric coefficient is 1–10 V, in this work we choose *f*_31_ = 3.9526 V, and the flexoelectric coefficient is computed as $\mu_{1133}=f_{31}\times \kappa_{33}=3.9526\times 1.265\times 10^{\left(-8\right)}=5\times 10^{\left(-8\right)}C/m$.

Firstly, in Table 1, to verify the present formulation in the two-phase framework, the first vibrational frequency ratio of a nanobeam with flexoelectric effect is obtained by eliminating the FG base, and compared the results to those from [42]. In this table, the other constants are ξ=1 and $L=40\;h_{d}$. Additionally, the frequency ratio in the reference is defined as follows.
(35)$${\bar{F}}_{r}=\frac{\;\mathrm{Frequency}\;\mathrm{of}\;\mathrm{the}\;\mathrm{nanobeam}\;\mathrm{without}\;\mathrm{flexoelectricity}}{\;\mathrm{Frequency}\;\mathrm{of}\;\mathrm{the}\;\mathrm{nanobeam}\;\mathrm{with}\;\mathrm{flexoelectricity}}$$

Next, to study the validity of the two-phase theory formulation, in Table 2, the dimensionless frequencies of a nanobeam with omitting the effect of piezoelectricity and flexoelectricity are tabulated and compared with those of Ref. [26]. The other constant which plays a role in determining the values is k=0.05 L. The results are presented for simply supported (SS) and clamped-free (CF) end conditions. It should be noted that the exact results are obtained using the equations and solution procedure presented in the reference. However, the result which is entitled from Ref. [26] is extracted approximately from the figure presented in this article.

As can be seen from these two tables, the close agreement between the results acquired through the presented formulation and solution method confirms the credibility of these two in exploring the vibration behavior of a two-layered nanobeam, including an FG base and a dielectric layer with the flexoelectricity and piezoelectricity. In addition, it should be mentioned that two-phase theory can have good agreement with the results of molecular dynamic models [25], making it a good tool to investigate small-sized structures. Additionally, by setting ξ=1 the results associated with the classic continuum are obtained, showing the validity of the current continuum theory.

Here, the frequency ratio, which is defined below, for different nanobeam lengths and boundary conditions are tabulated in Table 3. The other constants used in this table are k=0.05 L, ξ=0.1, *n* = 0, and $h_{d}=0.005\;L\;$.
(36)$$F_{r}=\frac{\;\mathrm{Frequency}\;\mathrm{of}\;\mathrm{the}\;\mathrm{nanobeam}\;\mathrm{with}\;\mathrm{flexoelectricity}}{\;\mathrm{Frequency}\;\mathrm{of}\;\mathrm{the}\;\mathrm{nanobeam}\;\mathrm{without}\;\mathrm{flexoelectricity}}$$

It can be understood from this table that the impact of flexoelectricity on the vibrational frequency of a unimorph system can be intensified by reducing the length of the nanobeam, so much so that the highest frequency ratio, regardless of the boundary condition, occurs in the cases with the lowest length.

Now, the impact of slenderness related to the FG base of the two-layered nanobeam with an FG layer and a dielectric layer on its frequency ratio is examined for various FG index numbers in Figure 3. Additionally, other constants are k=0.05 L, ξ=0.1, and $h_{d}=0.005\;L\;$.

Figure 3 exhibits that intensifying the base layer’s thickness ratio diminishes the value of the frequency ratio of the nanobeam. In other words, the thicker base causes the effect of flexoelectricity to reduce. Additionally, it is worth noting that the cases with a higher FG index can possess a higher frequency ratio, which means that the flexoelectricity is higher if the FG power index is higher. Further, the higher the FG power index is, the higher the metal phase’s contribution in the base is. It can be seen that the frequency ratio is dominated by the base plate and thickness ratio of the beam rather than end conditions.

Next, the impact of considering flexoelectricity in the current model, specifically small-scale, on the vibration response of the system, is shown in Figure 4. In this figure, the non-dimensional vibration frequency of the beam is plotted against the value of the thickness ratio of the base for three models which consider flexoelectricity and piezoelectricity, only flexoelectricity, and only piezoelectricity. Additionally, the non-dimensional vibration frequency is obtained through the following equation.
(37)ω¯=ωL2I¯1E¯I¯effE¯I¯eff=−b∫−h2h2Ecm z2dz−b∫h2h2+hdc11z2dzI¯1=b∫−h/2h/2ρcmdz+b∫h/2h/2+hdρddz

Similar to the previous figure, it can be seen that the effect of flexoelectricity rises with reducing the thickness of the nanobeam. Additionally, it can be concluded that, as the model with flexoelectricity and piezoelectricity yields a similar frequency to the one that only considers flexoelectricity, and hence, flexoelectricity is considered the dominant electromechanical coupling on small scales.

Here, Figure 5 deals with the impact of the thickness ratio of the dielectric layer with respect to the base layer’s thickness on the vibrational frequency of the system. Additionally, other constants in this figure are k=0.05 L, ξ=0.1, and h=0.01 L . The quite interesting results indicate that increasing the dielectric layer’s thickness causes the system’s vibrational frequency to increase with a high slope when the thickness ratio is far from 0.5. By increasing the value, the increment rate diminishes to a point in which the vibration frequency slightly decreases.

Now, the effect of two-phase elasticity parameters is investigated on the non-dimensional vibration of a two-layered nanobeam, including an FG base and a dielectric layer. To this aim, in Figure 6 and Figure 7, respectively, the influence of ξ and k/L on the fundamental vibration frequency of the two-layered system was analyzed for different end conditions and values for the FG index. Additionally, the other parameters that can affect the results are $L=50\;\left(\mathrm{nm}\right)$, $h_{d}=0.005\;L\;$, and h=0.01 L . Additionally, in Figure 6, the arrowheads show the dimensionless vibration frequency obtained by the differential form of purely nonlocal. Further, in Figure 8, the non-dimensional frequencies are obtained according to the differential form of purely nonlocal and are plotted against various nonlocality.

In addition, it can be understood that escalating ξ can cause the vibrational frequency of the nanobeam to increase, which is more observable in C-C and C-F end conditions. This means that the frequency of the unimorph is lower in the cases that nonlocal forces have bigger impacts and lower values of ξ. Additionally, as previously stated, the nanobeam in which the FG base has a higher index value can have a higher frequency, despite the boundary conditions and ξ values. FG index shows the function based on which the FG between two surface layers of ceramic and metal forms. Furthermore, it should be mentioned that, according to the purely nonlocal model, by decreasing the value of ξ toward 0, the vibration frequency of the system in the two-phase framework should lead to the values obtained by the differential form of purely nonlocal. Thus, as it can be seen in this figure, except for the cases with SS end conditions, the results of the two-phase model are significantly different from those attained via differential form of purely nonlocal elasticity, showing the inconsistency of this model due to the lack of additional end condition in the transformation from integral to differential model. In addition, it should be mentioned that studies have shown that not all of the forces between the particle of small-scale structures are purely nonlocal, and it is more a combination of local and nonlocal forces [53]. Thus, the two-phase theory is a more reliable mathematical model in order to investigate small-scale structures. Further, the value of ξ depends on the various geometrical parameters of the structure, which should be determined in every case specifically.

Interesting results of Figure 7 show the importance of using two-phase elasticity in order to study the small-scale structures with consideration to the piezoelectric and flexoelectric coupling, as by increasing the nonlocality of nanobeam, the value of frequency diminishes. In other words, by comparing the results of Figure 7 and Figure 8, decreasing frequency shows the softening effect which the nonlocal integral elasticity proposed and differential form of purely nonlocal theory is insufficient to capture, specifically in the C-F end condition. Additionally, similar to the previous figure, this softening effect can be more observable in C-F, C-C, and C-S end conditions compared to S-S one. The only end condition for which differential form of purely nonlocal can produce a comparatively accurate result is S-S, as the effect of additional boundary conditions is the least in this condition.

Here, in Figure 9, the impact of the flexoelectric coefficient on the vibrational frequencies of a unimorph system containing an FG base in addition to a dielectric layer is examined. In these figures, the variation of vibration frequencies of the system is plotted against various values for the flexoelectric coefficient for different nonlocality values. Additionally, the other constants in this figure are ξ=0.1, n=1, $L=50\;\left(\mathrm{nm}\right)$, $h_{d}=0.005\;L\;$, and h=0.01 L .

The results from this figure indicate that the vibrational frequency of the unimorph system is higher, provided that the flexoelectric coefficient associated with the dielectric layer possesses a higher positive or negative value. Additionally, the softening effect due to nonlocality can be observed in this figure, as the vibrational frequency is lower, despite the value of $\mu_{31}$, for beams with higher nonlocality.

Now, by putting a harmonic load with an excitation frequency of 0.85 f_1_ to 1.15 f_1_—in which f_1_ represents the fundamental vibration frequency of the system—the displacement response and output voltage of the system are obtained. Additionally, ${\bar{f}}_{0}=2\times 10^{-9}\frac{N}{m}$ in a condition that $L=50\;\left(\mathrm{nm}\right)$, $h_{d}=0.005\;L\;$, and h=0.01 L . Additionally, it should be noted that the deflection FRFs are presented for the endpoint in CF and the middle point for the CC boundary conditions. Next, the impact of different parameters on the frequency response function (FRF) associated with nanobeam’s deflection and output voltage are investigated.

Firstly, the deflection FRF for two types of boundary conditions for three different models are presented in Figure 10. The three models used are the beams considering only piezoelectric coupling, flexoelectric coupling, and a combination of both electromechanical couplings. It can be seen that flexoelectricity is the dominant form of electromechanical coupling, as the result related to the case that considers both of the couplings has a similar outcome to the one only considering the flexoelectricity. Also, as previously shown, the vibrational frequency of the system with flexoelectricity has a higher resonant frequency than the one in which piezoelectricity is the only considered form of electromechanical coupling. Therefore, these results indicate that in order to reach a better model for dielectric materials on a small scale, flexoelectric coupling must be considered the dominant form of electromechanical coupling.

Additionally, in Figure 11 and Figure 12, respectively, the impact of the base’s slenderness on the displacement and voltage FRF of the unimorph system with two boundary conditions is examined. The interesting results in these figures indicate that intensifying h/L of the nanobeam can cause the deflection of the beam in addition to the output voltage to diminish, regardless of the boundary condition. However, the deflection and output voltage are higher provided that the softer end condition is utilized. The other notable result is that the output voltage and displacement peak occur in higher frequencies by decreasing h/L.

Next, in Figure 13 and Figure 14, the displacement and voltage FRFs are plotted for different nonlocality and boundary conditions. In addition, other constants are equal to ξ=0.1, n=1, $L=50\;\left(\mathrm{nm}\right)$, $h_{d}=0.005\;L\;$, and h=0.01 L . It can be seen that the peaks related to deflection FRF and voltage FRF are moving towards lower frequencies due to softening effect of nonlocality. Therefore, as the cases with a higher nonlocality are softer, their deflection corresponding to these cases is higher, meaning they possess higher energy and can produce higher voltages. Similar to the previous figure, the output voltage and deflection are higher in softer boundary conditions.

Lastly, the displacement and voltage FRFs for different ξ are plotted in Figure 15 and Figure 16. In addition, the other constants are as follows k/L=0.05, n=1, $L=50\;\left(\mathrm{nm}\right)$, $h_{d}=0.005\;L\;$, and h=0.01 L . As expected, increasing the value leads to increasing the system’s natural frequency, moving the deflection and voltage peak in FRFs towards higher frequencies. Consequently, as the system is less nonlocal by intensifying ξ, the system is stiffer and produces less voltage, regardless of the boundary conditions.

## 5. Conclusions

This research presents a study on the frequency response of a nanobeam made of a FG base along with a dielectric layer considering both flexoelectric and piezoelectric couplings. The effect of the small size is considered by means of a paradox-free elasticity named two-phase local/nonlocal theory. Then, the equations of motions and end conditions are extracted using the Euler–Bernoulli beam theory, variational energy method, and GDQM. The results of this study are validated by employing other published articles. Finally, a parametric investigation is presented, the most important results of which are:

The smaller the nanobeams are, the dominant effect of flexoelectricity over piezoelctricty can be observed more clearly.

The lower values of ξ the system are closer to being purely nonlocal, causing the frequency to diminish. On the other hand, increasing ξ and h/L leads to a reduction in the peak values of voltage and displacement FRFs.

By intensifying the FG index, the vibrational frequency as well as the flexoelectric effect of the dielectric layer can be increased.

The higher the nonlocality, the higher the voltage and displacement FRFs peaks.

The two phase instead of differential nonlocal possess no paradoxes, making a reliable model by which the behavior of MEMS and NEMS can be studied. These findings suggest promising applications in nanoenergy transduction, nanogenerator, nanosensing and nanoactuation.

## Figures and Tables

**Figure 1 materials-16-03485-f001:**
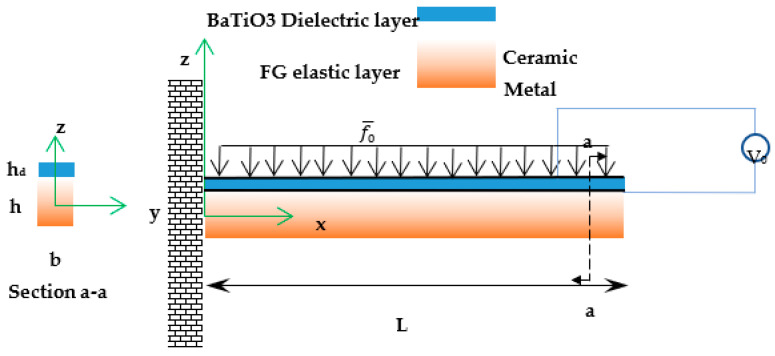
Schematic view of a two-layered piezoelectric nanobeam.

**Figure 3 materials-16-03485-f003:**
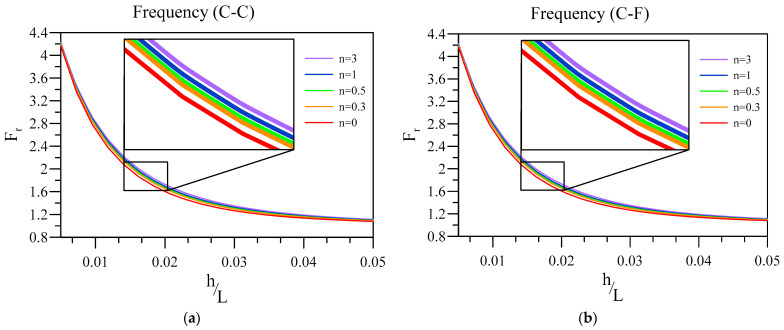
Variation of frequency ratio of a two-layered nanobeam against h/L for various FG index numbers. (**a**) C-C and (**b**) C-F boundary conditions.

**Figure 4 materials-16-03485-f004:**
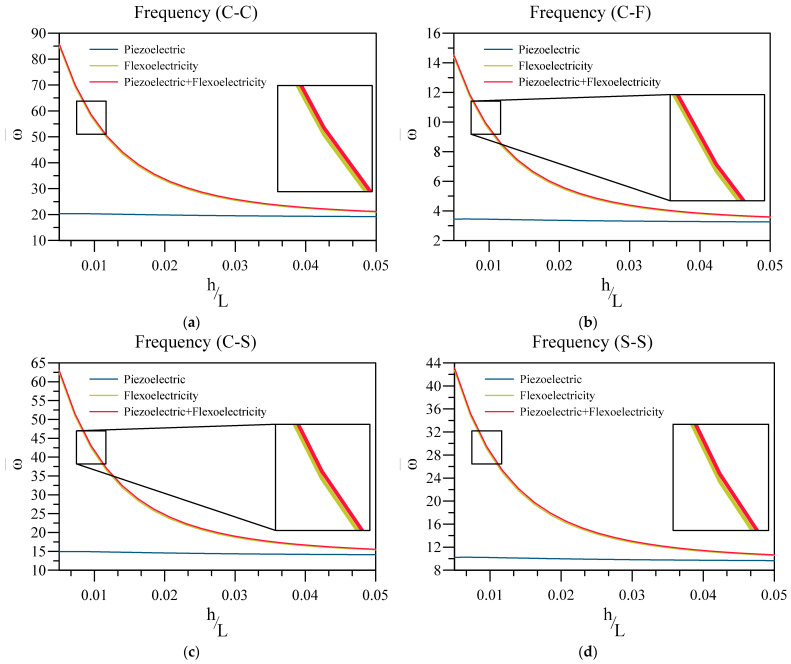
Variation of fundamental vibration frequency of two-layered nanobeams against h/L for (**a**) C-C, (**b**) C-F, (**c**) C-S, and (**d**) SS boundary conditions and three modulus.

**Figure 5 materials-16-03485-f005:**
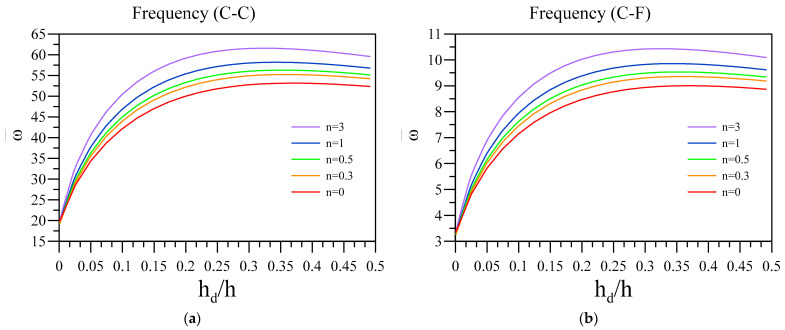
Variation of fundamental vibration frequency of two-layered nanobeams against h_d_/h for various *n* and (**a**) C-C and (**b**) C-F boundary conditions.

**Figure 6 materials-16-03485-f006:**
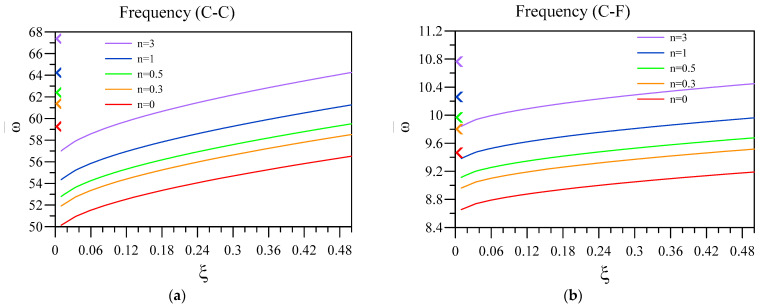
Variation of fundamental vibration frequency of two-layered nanobeams against ξ for (**a**) C-C, (**b**) C-F, (**c**) C-S, and (**d**) SS boundary conditions.

**Figure 7 materials-16-03485-f007:**
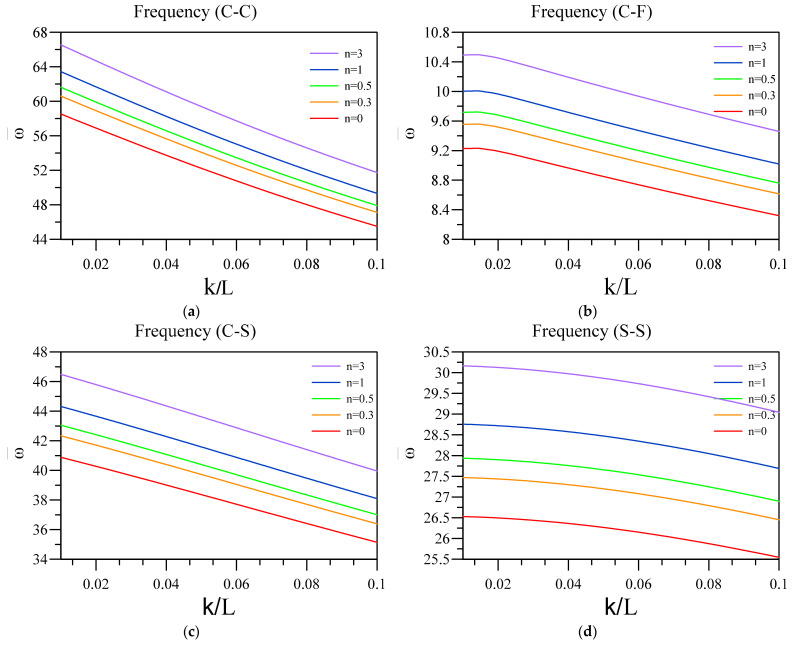
Variation of fundamental vibration frequency of two-layered nanobeams against k/L for (**a**) C-C, (**b**) C-F, (**c**) C-S, and (**d**) SS boundary conditions.

**Figure 8 materials-16-03485-f008:**
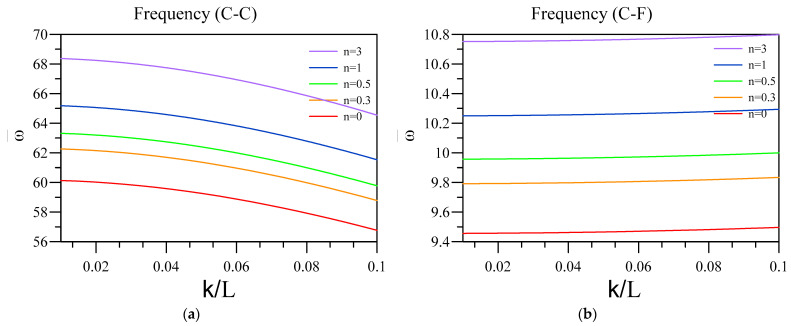
Variation of fundamental vibration frequency of two-layered nanobeams obtained via differential nonlocal against k/L for (**a**) C-C and (**b**) C-F boundary conditions.

**Figure 9 materials-16-03485-f009:**
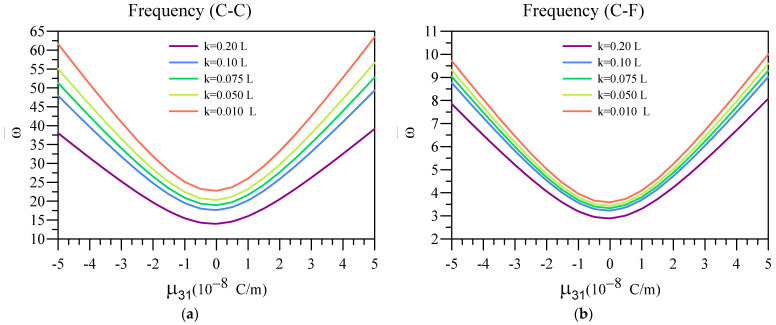
Variation of fundamental vibration frequency of two-layered nanobeams against $\mu_{31}$ for (**a**) C-C, (**b**) C-F, (**c**) C-S, and (**d**) SS boundary conditions.

**Figure 10 materials-16-03485-f010:**
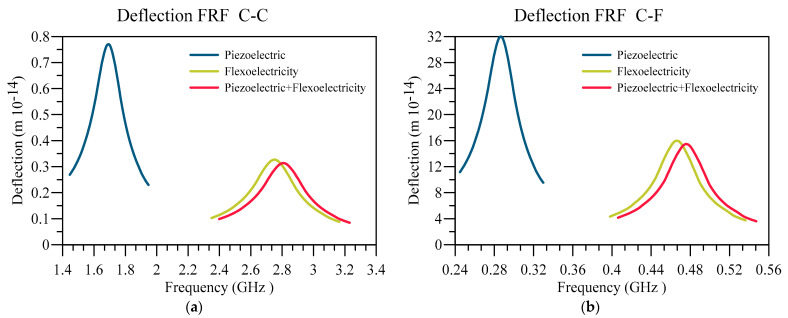
Displacement FRF for different beam types and (**a**) C-C and (**b**) C-F boundary conditions.

**Figure 11 materials-16-03485-f011:**
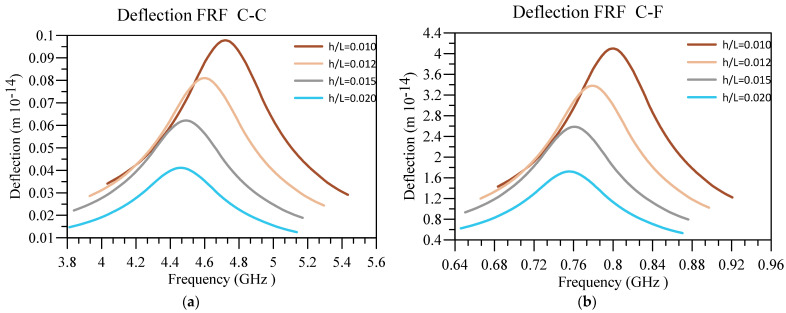
Displacement FRF for different values of h/L and (**a**) C-C and (**b**) C-F boundary conditions.

**Figure 12 materials-16-03485-f012:**
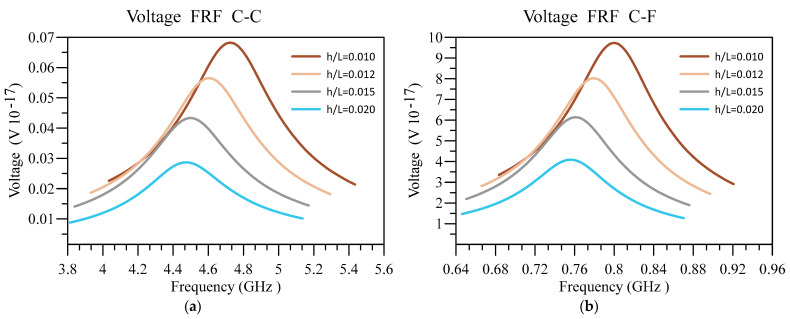
Voltage FRF for different values of h/L and (**a**) C-C and (**b**) C-F boundary conditions.

**Figure 13 materials-16-03485-f013:**
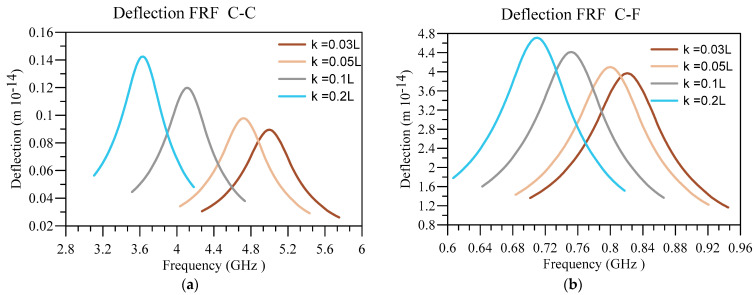
Displacement FRF for different values of κ/L and (**a**) C-C and (**b**) C-F boundary conditions.

**Figure 14 materials-16-03485-f014:**
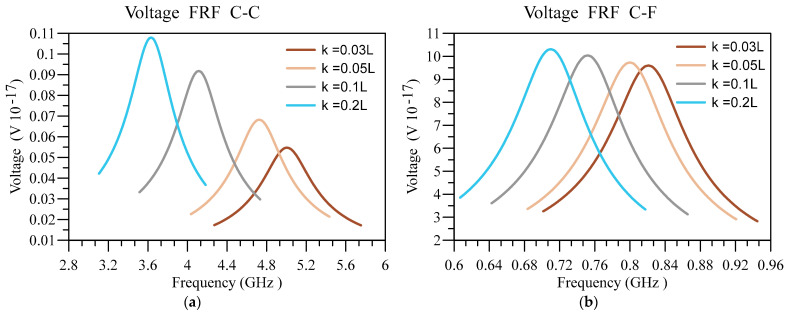
Voltage FRF for different values of κ/L and (**a**) C-C and (**b**) C-F boundary conditions.

**Figure 15 materials-16-03485-f015:**
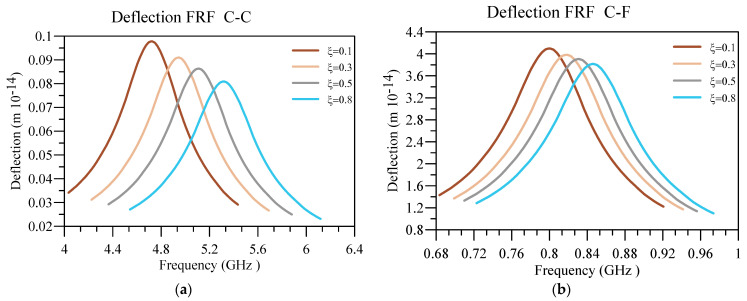
Displacement FRF for different values ξ of h/L and (**a**) C-C and (**b**) C-F boundary conditions.

**Figure 16 materials-16-03485-f016:**
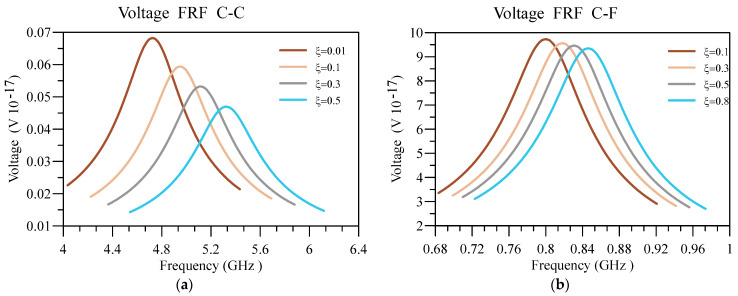
Voltage FRF for different values ξ of h/L and (**a**) C-C and (**b**) C-F boundary conditions.

**Table 1 materials-16-03485-t001:** The frequency ratio of the piezoelectric nanobeam with respect to the flexoelectricity.

	$h_{d}\left(\mathrm{nm}\right)$				
	5	10	15	20	25
Present results	0.83504	0.96573	0.98432	0.99109	0.99427
Ref. [42] *	0.83419	0.96318	0.98386	0.99092	0.99420
Error %	0.10	0.26	0.04	0.01	0.01

* It should be noted that the reference results are approximately obtained from a figure.

**Table 2 materials-16-03485-t002:** The first two dimensionless frequencies of a nanobeam with omitting the piezoelectricity as well as flexoelectricity for SS and CF end conditions, and k/L=0.05.

		Modes							
		1st				2nd			
BCs	ξ	Ref. [26] *	Exact [26] **	Present	Error %	Ref. [26] *	Exact [26] **	Present	Error %
SS	1.00	9.8696	9.8696	9.8696	0.00	39.4784	39.4784	39.4784	0.00
	0.50	9.8192	9.8148	9.8145	0.01	38.6493	38.6502	38.6502	0.00
	0.10	9.7610	9.7671	9.7671	0.00	37.8993	37.9218	37.9218	0.00
	0.05	-	9.7601	9.7601	0.00	-	37.8150	37.8150	0.00
	0.01	-	9.7533	9.7533	0.00	-	37.7125	37.7125	0.00
CF	1.00	3.5160	3.5160	3.5160	0.00	22.0344	22.0345	22.0345	0.00
	0.50	3.4105	3.4173	3.4173	0.00	21.2628	21.2626	21.2626	0.00
	0.10	3.2874	3.2908	3.2908	0.00	20.2713	20.3538	20.3538	0.00
	0.05	-	3.2617	3.2617	0.00	-	20.1592	20.1592	0.00
	0.01	-	3.2233	3.2233	0.00	-	19.9118	19.9118	0.00

* Ref. [26] values are attained from figures in this reference. ** The exact values are extracted by solving the method which is given in Ref. [26].

**Table 3 materials-16-03485-t003:** The frequency ratio of the piezoelectric nanobeam with considering the flexoelectricity.

BC	$L\;\left(\mathrm{nm}\right)$					
	20	50	100	200	500	1000
S-S	1.19167	1.03937	1.01247	1.00438	1.0013	1.00058
C-F	1.19167	1.03937	1.01247	1.00438	1.0013	1.00058

## Data Availability

Not applicable.

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
