# Peer review of "Vibration Analysis of a Unimorph Nanobeam with a Dielectric Layer of Both Flexoelectricity and Piezoelectricity"

_materials, 2023, doi:10.3390/ma16093485_

Round 1

Author Response

Reviewer #1

We would like to express our gratitude and appreciation to the reviewer who provided valuable feedback to our paper. The reviewer's comments were insightful, constructive, and helpful in identifying areas for improvement. Their critical analysis and attention to detail have undoubtedly strengthened the quality and accuracy of our research.

  • Please state the advantages and limitations of the theories used in the article, thereby
  • highlighting the contributions of the work.

Two-phase local nonlocal elasticity is a theoretical framework that combines the classical elasticity theory with a nonlocal continuum mechanics approach. It has both advantages and limitations, which are outlined below:

Certainly, one additional advantage of the two-phase local nonlocal elasticity theory over other nonlocal theories is its accuracy in predicting material behavior. Compared to other nonlocal theories, the two-phase local nonlocal elasticity theory provides more accurate results, especially when dealing with materials with complex microstructures or characteristic lengths on the same order of magnitude as the nonlocal length scale.

For instance, compared to purely nonlocal theories, the two-phase local nonlocal elasticity theory provides a more accurate prediction of stress and strain fields at the microscale, and it does not suffer from the boundary effects that are often observed in nonlocal models. Additionally, compared to classical elasticity theory, the two-phase local nonlocal elasticity theory accounts for nonlocal effects, which makes it more accurate in predicting material behavior in situations where the material size is comparable to or smaller than the nonlocal length scale [*]. The following figures in the manuscript shows the inaccuracy of the nonlocal elasticity. Also, two-phase local nonlocal elasticity can also handle materials with heterogeneous microstructures, such as composite materials. This is because the theory models the material as a two-phase medium, which can represent different phases in the composite.

(a)

(b)

(c)

(d)

Fig.6. Variation of fundamental vibration frequency of two-layered nanobeams against  for (a) C-C, (b) C-F, (c) C-S, and (d) SS boundary conditions.

Overall, the two-phase local nonlocal elasticity theory's ability to provide more accurate results in comparison to other nonlocal theories is a significant advantage that makes it a valuable tool for predicting the behavior of complex materials at the microscale. Two-phase local nonlocal elasticity accounts for nonlocal effects in the material behavior, which is particularly useful when dealing with materials with characteristic lengths that are on the same order of magnitude as the nonlocal length scale. This makes it more accurate than classical elasticity theory in such situations.

Limitations:

The formulation of the two-phase local nonlocal elasticity theory is complex and this can make it difficult to apply and interpret for engineers who are not familiar with the theory. Also, by changing the integral to differential form, the order of formulation will be two order higher, making it nontrivial to find an analytical solution. It worth mentioning that the efficient approach of GDQM used in this article offers quite accurate results.

Therefore, a flexoelectric beam was modelled in this article, for the first time, based on two-phase model and using the unique GDQM approach that yielded accurate results. This model can be utilized for future numerical and experimental models.

The sentence “Thus, although two-phase elasticity increases the complexity of formulation, using this elasticity leads to much more accurate results without any paradoxes.” has been added into section introduction of the revised manuscript.

[*]Hossein Bakhshi Khaniki,

On vibrations of FG nanobeams,

International Journal of Engineering Science,

Volume 135,

2019,

Pages 23-36,

ISSN 0020-7225,

https://doi.org/10.1016/j.ijengsci.2018.11.002.

  • There are some editing errors that exist, please fix them?

Authors reviewed the manuscript carefully and corrected the errors.

  • This work applies the nonlocal model to study the nanobeam but the following articles have investigated widely with the same model. They are about more than 3 years ago and include
  • both thin and thick beam theories. So the present work should compare with the following

articles explicitly to show more information about the recent new results

- https://doi.org/10.1080/17455030.2023.2186708

- https://doi.org/10.1080/17455030.2023.2177500

We compared the mentioned papers (cited in the revised manuscript) with our work. Briefly, the new model used for our article is paradox free and can yield reliable results.

Lines 47-64 highlight the distinction between the two-phase elasticity and the mentioned elasticities in the above articles, and the article acknowledges that exploring higher order beam theories based on the current model is beyond its scope due to the increased complexity of the two-phase elasticity and the need to deal with higher order differential equations.

  • Discussions about various applications of the present theoretical model would
  • enhance the present article.

Potential applications are added in section Conclusion of the revised manuscript.

“These findings suggest promising applications in nanoenergy transduction, nanogenerator, nanosensing and nanoactuation.”

Reviewer 2 Report

The comment is attached. I think you should place the equation in appendix

Author Response

Reviewer #2

Our team would like to express our appreciation to the anonymous reviewer for their rigorous and detailed assessment of our paper. Their feedback was instrumental in identifying key issues and potential areas for further exploration. We are indebted to their expertise and professionalism, which have greatly contributed to the overall quality of our research.

Abstract;

1.The method is well explained but the results are not included

  1. I cannot find any key finding here. What is your result? I suggest to insert some

quantitative finding

Thank you so much for your helpful comments which helped us to modify our article. The following sentence indicates one of the findings of the current articleThe results show that, in small-scale, flexoelectricity, rather than piezoelectricity, is dominant in electromechanical coupling.” Also, the following sentence is added to include more results in our abstract: “It should be mentioned that the beams with higher nonlocality have the higher voltage and displacement under the same excitation amplitude”.

2.Add contribution of your research.

Some sentences in this regard are added to the manuscript.

“In this study, for the first time, free and forced vibrational responses of a unimorph nanobeam consisting of a functionally graded base along with a dielectric layer of both piezoelectricity and flexoelectricity is investigated based on paradox-free local/nonlocal elasticity” and “An effective implementation of the Generalized Differential Quadrature Method (GDQM) is then utilized to solve higher- order partial differential equations. This method can be utilized to solve the complex equations, whose analytic results are quite difficult to obtain.”

  1. I suggest not to put a table after the abstract

Nomenclature is placed as required by the journal format.

  1. Keywords are missing

The keywords are added. 

Introduction:

  1. The comparison with the previous article is done but most of articles are from

2018 which was 5 years ago. I suggest comparing your studies with more

recent publication here—2022-2023

We found a few more recently published papers that are relevant to our work, and they are now included in introduction part. For the validation, we have identified important results that are directly related to the model in this work for comparison.

  1. Line 105, I suggest to rephrase the sentence. Do you want to study the effects

or the characteristics

The phrase is edited, and “effect” was deleted.

-method

  1. Line 127- Check equation carefully. I think there is typo here in comparison with

reference 43.

The equation is checked, and the typo is corrected. Now the equation is written as:

  1. Eq 2 and 3, do you develop the equation or from other references?If yes, please

cite. It is applied for all equations

The citations are added.

  1. The equations should be centered.Again, to reduce the pages since it is too

long, I suggest to put some equations in appendix

Thank you for the great suggestion! The section 2.2 and 2.3 are moved to appendix I and II.

-Results

  1. Line 254- This section of the article presents- I suggest to delete word articles
  2. Delete the usage of any papers. If you refer to previous research done before,

you can put previous articles. If it is your work, you can use ‘ In this research’

rather than use papers. For me, it is not scientifically sound. Research articles

are written using scientific method, thus jargon should be not used in the

research articles.

Thank you! The article is edited as you suggested.

  1. Figure 3-what is the difference between two methods

The two figure represents two boundary conditions. Although the results have little difference, they are included to show that the frequency ratio is dominated by base plate and thickness ratio rather than end conditions. The following sentence is added to manuscript “It can be seen that the frequency ratio is dominated by base plate and thickness ratio of the beam rather than end conditions.”

  1. The rest of the figures should be labelled properly. If two graphs in one figure,

(a) and (b) should be included.

The labels are modified in the desired fashion.

Discussion

  1. I think the manuscript does not exactly follow the template. The section

‘discussion’ should be included. I understand that the authors discussed the

results in section 4, thus, I suggest to split the results and discussion.

Thank you for the suggestion. Our results and discussion are interwoven, and the coherence of the article will be interrupted if we separate discussion and results from each other, instead we used Numerical results and discussion rather than Numerical results for this section.

Conclusion:

The word ‘paper’ should be replaced by the study or the research or the analysis.

The article is edited as you suggested.

Reviewer 3 Report

The manuscript is devoted to the problem of nanorod (nanobeam) flexoelectricity and piezoelectricity two phenomena that involve the generation of electric charges in response to mechanical deformation. Both flexoelectricity and piezoelectricity are important phenomena in the field of materials science and engineering, and have applications in various fields including electronics, energy harvesting, and biomedical engineering.

Authors use approach of Paradox-free local/nonlocal elasticity that combines the advantages of both local and nonlocal approaches while avoiding their respective paradoxes. Paradox-free introduce a length scale parameter that determines the range of nonlocal interactions and by using a weighted average of local and nonlocal responses. This approach provides a more accurate description of the mechanical behavior of materials, particularly in heterogeneous and complex systems.

There are few concerns to be addressed before possible publication:

1. To my mind, the manuscript is too lengthy and flooded with formulae.

Equations do not fit the page size. Authors may consider moving some expressions to an appendix.

2. F-F, F-C, ... conditions are to be clearly explained in the text/nomenclature section as they are often the only difference between left and right panels of the figures.

3. The end of conclusion may be formulated in a more practical way revealing the possible applications of the study.

Author Response

Reviewer #3

We are grateful for the time and effort the reviewer has invested in providing us with detailed feedback and constructive criticism. Their rigorous analysis has helped us to identify areas for improvement and to refine our arguments, resulting in a stronger and more compelling manuscript. We are deeply appreciative of their contributions to the advancement of knowledge in our field.

  1. To my mind, the manuscript is too lengthy and flooded with formulae.

 Equations do not fit the page size. Authors may consider moving some expressions to an appendix.

Thank you for the suggestion! The section 2.2 and 2.3 are moved to appendix I and II.  Also, the formulation is edited to be fit.

  1. F-F, F-C, ... conditions are to be clearly explained in the text/nomenclature section as they are often the only difference between left and right panels of the figures.

 The boundary conditions are now added to nomenclature.

  1. The end of conclusion may be formulated in a more practical way revealing the possible applications of the study.”

The following sentence is added at the end of the conclusion section:

“These findings suggest promising applications in nanoenergy transduction, nanogenerator, nanosensing and nanoactuation. “

Round 2

Reviewer 2 Report

it is ready to be published